# Design and Rationale of the Sevoflurane for Sedation in Acute Respiratory Distress Syndrome (SESAR) Randomized Controlled Trial

**DOI:** 10.3390/jcm11102796

**Published:** 2022-05-16

**Authors:** Raiko Blondonnet, Laure-Anne Simand, Perine Vidal, Lucile Borao, Nathalie Bourguignon, Dominique Morand, Lise Bernard, Laurence Roszyk, Jules Audard, Thomas Godet, Antoine Monsel, Marc Garnier, Christophe Quesnel, Jean-Etienne Bazin, Vincent Sapin, Julie A. Bastarache, Lorraine B. Ware, Christopher G. Hughes, Pratik P. Pandharipande, E. Wesley Ely, Emmanuel Futier, Bruno Pereira, Jean-Michel Constantin, Matthieu Jabaudon

**Affiliations:** 1Department of Perioperative Medicine, CHU Clermont-Ferrand, 63000 Clermont-Ferrand, France; rblondonnet@chu-clermontferrand.fr (R.B.); lasimand@chu-clermontferrand.fr (L.-A.S.); p_vidal@chu-clermontferrand.fr (P.V.); lborao@chu-clermontferrand.fr (L.B.); n_bourguignon@chu-clermontferrand.fr (N.B.); dmorand@chu-clermontferrand.fr (D.M.); jaudard@chu-clermontferrand.fr (J.A.); tgodet@chu-clermontferrand.fr (T.G.); jebazin@chu-clermontferrand.fr (J.-E.B.); efutier@chu-clermontferrand.fr (E.F.); 2iGReD, Université Clermont Auvergne, CNRS, INSERM, 63000 Clermont-Ferrand, France; lroszyk@chu-clermontferrand.fr (L.R.); vsapin@chu-clermontferrand.fr (V.S.); 3Department of Clinical Research and Temporary Authorization, CHU Clermont-Ferrand, 63000 Clermont-Ferrand, France; l_bernard@chu-clermontferrand.fr; 4Department of Medical Biochemistry and Molecular Genetics, CHU Clermont-Ferrand, 63000 Clermont-Ferrand, France; 5Department of Anesthesiology and Critical Care, GRC 29, DMU DREAM, Pitié-Salpêtrière Hospital, Sorbonne University, Assistance Publique-Hôpitaux de Paris, 75013 Paris, France; antoine.monsel@aphp.fr (A.M.); jean-michel.constantin@aphp.fr (J.-M.C.); 6Department of Anesthesiology and Critical Care Medicine, DMU DREAM, Saint-Antoine University Hospital, Sorbonne University, Assistance Publique-Hôpitaux de Paris, 75012 Paris, France; marc.garnier@aphp.fr; 7Department of Anesthesiology and Critical Care Medicine, DMU DREAM, Tenon University Hospital, Sorbonne University, Assistance Publique-Hôpitaux de Paris, 75020 Paris, France; christophe.quesnel@aphp.fr; 8Division of Allergy, Pulmonary, and Critical Care Medicine, Department of Medicine, Vanderbilt University Medical Center, Nashville, TN 37232, USA; julie.bastarache@vumc.org (J.A.B.); lorraine.ware@vumc.org (L.B.W.); wes.ely@vumc.org (E.W.E.); 9Department of Cell and Developmental Biology, Vanderbilt University, Nashville, TN 37232, USA; 10Department of Pathology, Microbiology, and Immunology, Vanderbilt University Medical Center, Nashville, TN 37232, USA; 11Division of Anesthesiology Critical Care Medicine, Department of Anesthesiology, Vanderbilt University Medical Center, Nashville, TN 37232, USA; christopher.hughes@vumc.org (C.G.H.); pratik.pandharipande@vumc.org (P.P.P.); 12Critical Illness, Brain Dysfunction, and Survivorship Center, Vanderbilt University Medical Center, Nashville, TN 37203, USA; 13Anesthesia Service, Department of Veterans Affairs Medical Center, Tennessee Valley Healthcare System, Nashville, TN 37212, USA; 14Geriatric Research, Education and Clinical Center, Department of Veterans Affairs Medical Center, Tennessee Valley Healthcare System, Nashville, TN 37212, USA; 15Biostatistics and Data Management Unit, Department of Clinical Research and Innovation (DRCI), CHU Clermont-Ferrand, 63000 Clermont-Ferrand, France; bpereira@chu-clermontferrand.fr

**Keywords:** acute respiratory distress syndrome, sevoflurane, inhaled sedation, clinical trial

## Abstract

Preclinical studies have shown that volatile anesthetics may have beneficial effects on injured lungs, and pilot clinical data support improved arterial oxygenation, attenuated inflammation, and decreased lung epithelial injury in patients with acute respiratory distress syndrome (ARDS) receiving inhaled sevoflurane compared to intravenous midazolam. Whether sevoflurane is effective in improving clinical outcomes among patients with ARDS is unknown, and the benefits and risks of inhaled sedation in ARDS require further evaluation. Here, we describe the SESAR (Sevoflurane for Sedation in ARDS) trial designed to address this question. SESAR is a two-arm, investigator-initiated, multicenter, prospective, randomized, stratified, parallel-group clinical trial with blinded outcome assessment designed to test the efficacy of sedation with sevoflurane compared to intravenous propofol in patients with moderate to severe ARDS. The primary outcome is the number of days alive and off the ventilator at 28 days, considering death as a competing event, and the key secondary outcome is 90 day survival. The planned enrollment is 700 adult participants at 37 French academic and non-academic centers. Safety and long-term outcomes will be evaluated, and biomarker measurements will help better understand mechanisms of action. The trial is funded by the French Ministry of Health, the European Society of Anaesthesiology, and Sedana Medical.

## 1. Introduction

Volatile anesthetics such as isoflurane or sevoflurane have long been used to provide general anesthesia in the operating room; however, inhaled sedation is emerging as an option to provide intensive care unit (ICU) sedation and a phase III multicenter randomized controlled trial found that isoflurane is non-inferior to propofol in maintaining targeted sedation levels in critically ill adult patients [1,2,3,4,5]. Preclinical studies have shown that inhaled sevoflurane improves gas exchange [6,7,8], reduces alveolar edema [8], and attenuates pulmonary and systemic inflammation [9,10] in experimental models of acute respiratory distress syndrome (ARDS).

In addition to their potential effects on mechanisms relevant to ARDS [11], findings from clinical studies in critically ill patients suggest that volatile anesthetics may provide superior awakening and extubation times in comparison with intravenous sedatives such as propofol and benzodiazepines [12,13]. In a previous pilot randomized controlled trial of patients with moderate to severe ARDS, inhaled sevoflurane, compared to intravenous midazolam, improved oxygenation and decreased inflammation and lung epithelial injury, as assessed by plasma and an alveolar soluble receptor for advanced glycation end-products (sRAGE), interleukin (IL)-1β, IL-6, IL-8, and tumor necrosis factor (TNF)-α [14]. In this study and others [15,16,17], sevoflurane inhalation through dedicated devices was well tolerated, with no major adverse effects [18]. This pilot trial was underpowered to evaluate mortality or other major clinical outcomes. Given the number of ICU patients with ARDS receiving sedation and the overall burden of ARDS on healthcare, especially since the coronavirus disease 2019 (COVID-19) pandemic, improving clinical outcomes through sedative choice would have important implications globally.

The phase III multicenter clinical trial, SESAR (Sevoflurane for Sedation in ARDS), was designed to examine the efficacy and safety of a strategy of inhaled sevoflurane sedation compared with a strategy of current intravenous sedation practice using propofol in patients with ARDS.

## 2. Materials and Methods

The complete trial protocol, as currently approved, is provided in Appendix A.

### 2.1. Objectives

We hypothesize that a strategy of inhaled sedation with sevoflurane may be more effective than current intravenous sedation practice in improving clinical outcomes in ARDS. We investigate our hypotheses through the following objectives:To examine the efficacy of inhaled sevoflurane versus intravenous propofol in improving a composite outcome of mortality and time off the ventilator at 28 days in ARDS.To evaluate the safety of inhaled sevoflurane in ARDS (clinical adverse events), to describe its effects on the duration of mechanical ventilation, organ dysfunction, the use of rescue procedures, ICU-acquired delirium, major clinical and long-term outcomes, and healthcare-related costs during ICU and hospital stay.To investigate the physiological and biological mechanisms of protection by inhaled sevoflurane in ARDS, if any, and their potential roles in heterogeneity of treatment effects.

### 2.2. Trial Design

The SESAR trial is an investigator-initiated, two-arm, parallel-group, randomized controlled trial with a blinded outcome assessment. The primary endpoint is the number of days alive and off the ventilator at 28 days (ventilator-free days through day 28, VFD28), thereby considering death as a competing event. The planned enrollment is 700 adult participants at 37 French clinical sites. The trial protocol was approved by an ethics committee (*Comité de Protection des Personnes Ile-de-France* 2, approval number 18.09.21.60651-RIPH1) and the French Medicine Agency (*Agence Nationale de Sécurité du Médicament et des Produits de Santé*, approval number MEDMSANAT-2019-09-00248). The trial has been designed in accordance with the Standard Protocol Items: Recommendations for Interventional Trials guidelines (Appendix A).

### 2.3. Eligibility Criteria and Exclusions

Adult patients under invasive mechanical ventilation with ARDS [19] and a PaO_2_/FiO_2_ <150 mmHg under a positive end-expiratory pressure (PEEP) ≥8 cmH_2_O are enrolled, such as in the *ARDS et Curarisation Systematique* and *Reevaluation of Systemic Early Neuromuscular Blockade* trials [20,21]. A SpO_2_/FiO_2_ that is equivalent to a PaO_2_/FiO_2_ <150 mmHg can be used when an arterial blood gas is not available [22]. Exclusion criteria include known pregnancy, suspected or proven intracranial hypertension, persistent bronchopleural fistula despite chest tube drainage, long QT syndrome at risk of arrhythmic events, a medical history of malignant hyperthermia, liver disease attributed to previous exposure to volatile anesthetics, hypersensitivity or anaphylactic reaction to sevoflurane, propofol or cisatracurium, current treatment with sevoflurane or extracorporeal membrane oxygenation (ECMO) at enrollment, enrollment in another interventional trial with direct impact on sedation or mechanical ventilation, or if their tidal volume of 6 mL/kg predicted body weight (PBW) is below 200 mL (as recommended by the manufacturer of the device used to deliver inhaled sevoflurane in the trial). As oxygenation may improve during the 24 h enrollment window, an exclusion criterion is a PaO_2_/FiO_2_ >200 mmHg (if available) after meeting inclusion criteria and before randomization. Full eligibility criteria are listed in Table 1.

Due to eligibility criteria, patients will not be able to provide informed consent at enrollment and the study protocol provides for a waiver of informed consent from the patient. In case the patient’s legally authorized representative cannot be reached during the 24 h time window for enrollment, the investigator can include the patient using an emergent consent procedure and deferred informed consent will be obtained as soon as possible from the participants. The CONSORT diagram of the SESAR trial is provided in Figure 1.

### 2.4. Randomization

All patients were randomized online (CSOnline, Clinsight) by local investigators within 24 h of meeting inclusion criteria on a 1:1 ratio for the two study arms. The randomization sequence was generated by minimization and stratified by the study center, the degree of ARDS severity (PaO_2_/FiO_2_ < 100 mmHg), the suspicion or presence of COVID-19, and the presence of shock (defined as intravenous infusion of vasoactive drugs) at enrollment.

### 2.5. Treatment Arms, Administration, and Standardization of Care

#### 2.5.1. Interventions

Study interventions are summarized in Figure 2.

#### 2.5.2. Study Arms: Intervention—Inhaled Sedation with Sevoflurane

Sevoflurane is vaporized via the miniaturized Anesthesia Conserving Device (Sedaconda ACD-S, Sedana Medical, Danderyd, Sweden), which is placed between the endotracheal tube and the Y-piece of the ventilator breathing circuit. The residual expired gas is scavenged following the manufacturer’s instructions using an active carbon filter (Flurabsorb, Sedana Medical, Danderyd, Sweden) [1]. As recommended by the manufacturer, Sedaconda ACD is replaced every 24 h and removed from the breathing circuit as soon as inhaled sedation is interrupted and for spontaneous breathing trials. Sevoflurane administration is interrupted, and the Sedaconda ACD-S removed from the breathing circuit, if severe acidemia (pH < 7.15) is present, in the absence of metabolic acidosis and despite tidal volume and/or respiratory rate increase, or if malignant hyperthermia or a bronchopleural fistula that is persistent despite drainage (to limit room exposure) develops. In these situations, sedation is switched to a strategy of intravenous propofol.

#### 2.5.3. Study Arms: Control—Intravenous Sedation with Propofol

Propofol is administered via continuous intravenous infusion, as routinely used in participating ICUs. Propofol is interrupted if propofol-related infusion syndrome develops; in these cases, management and the choice of sedative agent(s) is as per the treating clinicians.

#### 2.5.4. Common Strategies for Both Groups

Upon randomization, deep sedation is protocolized (Richmond Agitation-Sedation Scale (RASS) −4 to −5) and combined with the neuromuscular blockade (continuous infusion of cisatracurium besylate for a maximum of 48 h) in both groups. The neuromuscular blockade is continued until PaO_2_/FiO_2_ exceeds 150 mmHg for 4 h with FiO_2_ < 0.6 [21,23,24]; then, light sedation is targeted (RASS 0 to −1), with prompt sedation interruption whenever possible. In both arms, patients receive the allocated sedation strategy from randomization until sedation can be interrupted or until day 7, whichever occurs first. When sedation is required again within 7 days after randomization, the sedative agent to use is based on the randomization arm; after day 7, decisions on sedation are as per the treating clinicians. If the bispectral index (BIS^®^, Aspect Medical Systems Inc., Norwood, MA, USA) is available, the level of sedation under the neuromuscular blockade can be titrated and monitored using the BIS^®^, with a targeted value of 40–50 [14]. Pain management is conducted as per clinical teams, within a strategy of analgesia-first sedation, including frequent pain assessment with the behavioral pain scale (BPS); opioids agents, if needed, are those routinely used in participating centers.

Lung-protective ventilation is used in both arms [25]. Using volume-controlled ventilation, tidal volume is set at 6 mL/kg (+/− 2 mL/kg) of PBW [26], and PEEP is kept as high as possible without increasing the inspiratory plateau pressure above 28–30 cmH_2_O [27]. We allow deviation from the high PEEP strategy if there is clinical concern that the use of high PEEP may be worsening oxygenation, if hypotension and/or high inspiratory plateau pressure (>30 cmH_2_O) are present despite further tidal volume reduction and/or respiratory rate increase, if a study participant develops a pneumothorax, is at high risk for barotrauma, or as per the treating clinicians. Instrumental dead space of the respiratory circuit is reduced to the minimum in both arms. Whenever possible, we recommend sites to wait at least 12 h before proning for more than 12 h/day, one or multiple times as per the treating clinicians [23,26]. Fluid management during shock is unrestricted; however, in patients not in shock, a conservative fluid approach is recommended (Appendix A) [28]. If PaO_2_ ≥ 55 mmHg or SpO_2_ ≥ 88% with FiO_2_ of 1 cannot be maintained, clinicians may employ rescue procedures, chosen according to the practice at the clinical site, such as recruitment maneuvers, nitric oxide, ECMO, or neuromuscular blockade use after 48 h from randomization. Evidence-based recommendations for weaning from mechanical ventilation will be applied in both groups (Appendix A). All participating centers have existing protocols and order sets for routine sedation management, glucose control, septic shock resuscitation, deep venous thrombosis prophylaxis, and other aspects of background care.

#### 2.5.5. Standardization and Adherence

Although all participating critical care professionals are certified and allowed to manage ICU sedation, their previous use and expertise level of inhaled sevoflurane may vary [4]. An educational program is conducted prior to patient recruitment to ensure that all participating centers have sufficient training to ensure patient safety and reach study goals through online-based theoretical presentations and online and on-site practical training. The study sponsor (CHU Clermont-Ferrand) safeguards data quality monitoring via web-based data collection, monthly query reports, site visits, and structured data collection training. A steering committee supervises the trial.

### 2.6. Patient Timeline, Assessments, and Measures

Patients are assessed as described in the Time-Events schedule (Appendix A). Long-term outcomes are assessed on days 90 and 365 (Appendix A). It is not technically possible to mask the assigned sedation strategy to the treating clinicians, due to the nature of the intervention requiring specific equipment. However, patients are followed up by members of the research staff from each site who are unaware of the group allocation. Study data are collected into the electronic case report form (CSOnline, Clinsight, Paris, France) by trained research staff blinded to the allocation group.

### 2.7. Trial Outcome Measures

The trial outcome measures are summarized in Table 2.

#### 2.7.1. Primary Outcome

The primary outcome is the number of days alive and off invasive mechanical ventilation at 28 days (ventilator-free days at 28 days, VFD28), considering death as a competing event. VFD28 are defined as the number of days from the time of initiating unassisted breathing to day 28 after randomization, assuming survival for at least two consecutive calendar days after initiating unassisted breathing and continued unassisted breathing to day 28. A period of assisted breathing lasting less than 24 h and for the purpose of a surgical procedure does not count against the VFD calculation, and VFDs are zero if a patient dies prior to day 28.

#### 2.7.2. Key Secondary Outcomes

The key secondary outcome is 90 day survival.

#### 2.7.3. Secondary Outcomes

Other secondary outcomes are all-location, all-cause mortality at 7, 14, and 28 days, and all-cause hospital 28 day mortality.

#### 2.7.4. Exploratory Outcomes

Exploratory outcomes are VFD7, VFD14, organ failure-free days through day 7 [evaluating Sepsis-related Organ Failure Assessment (SOFA) scores [29]], ICU-free and hospital-free days through day 28, respiratory physiological measures (oxygenation index, PaO_2_/FiO_2_, PaCO_2_, arterial pH, PEEP, inspiratory plateau pressure, static compliance of the respiratory system, and ventilatory ratio, as defined as [minute ventilation (mL/min) × arterial PaCO_2_ (mmHg)]/[predicted body weight *×* 100 *×* 37.5] [30]), use of rescue procedures for refractory hypoxemia through day 28, ICU-acquired delirium through day 7 or ICU discharge or death (whichever comes first), long-term outcomes at 3 and 12 months (disability, health-related quality of life, self-rated health, pain-interference, post-traumatic stress-like symptoms, cognitive function, subsequent return to work, healthcare use, and location of residence), and healthcare-related costs during ICU stay and stay (secondary analysis). Instruments for ICU-acquired delirium and long-term outcome assessment are detailed in Appendix A.

#### 2.7.5. Exploratory Biological Outcomes

Exploratory biological outcomes are changes over time in plasma biomarkers including IL-8, soluble tumor necrosis factor-receptor 1, bicarbonate (hyperinflammatory ARDS phenotype [31]), IL-6 (ventilator-induced lung injury [25,32]), angiopoietin 2 (endothelial activation [33,34]), and sRAGE (alveolar epithelial injury [35,36]), in urine biomarkers including tissue inhibitor of metalloproteinase 2 and insulin-like growth factor binding protein 7 (acute kidney injury [37]), in plasma total fluoride and hexafluoroisopropanol (sevoflurane metabolism), in total protein within undiluted pulmonary edema fluid (alveolar fluid clearance [36]), in biomarker measurements in the fluid from heat moisture exchanger filters (control group) or Sedaconda ACD-S devices (intervention group) [38,39] and in the bronchoalveolar lavage fluid, and genetic analyses (Table 2).

#### 2.7.6. Safety Outcomes

Safety outcomes are hemodynamic (mean arterial pressure, dose of infused norepinephrine or other vasopressor, serum lactate level) and renal function (KDIGO criteria [40]) measures through day 7 and development of severe hypercapnic acidosis with arterial pH < 7.15 (Sedaconda ACD-S device), supraventricular tachycardia or new onset atrial fibrillation during the ICU stay, malignant hyperthermia (sevoflurane), propofol-related infusion syndrome (propofol), and bronchopleural fistula that is persistent despite drainage (to limit room exposure to sevoflurane) through day 7.

### 2.8. Laboratory Evaluations

In addition to laboratory studies obtained as per routine clinical care, plasma and urine samples will be collected, when available, at study entry and on days 1, 2, 4, 6, and 14 or ICU discharge (whichever occurs first) to investigate the biological effects of inhaled sevoflurane in ARDS. We will also collect whole blood samples at study entry and on day 2 for RNA and DNA studies. In selected centers, undiluted pulmonary edema fluid samples will be collected at study entry and 24 h later in 50 patients from each group; bronchoalveolar lavage fluid will be sampled in 25 intubated patients within 48 h from study entry and between day 4 and day 6; fluid from heat moisture exchanger filters and Sedaconda ACD-S devices will be collected at 24 h in 30 patients from each group.

#### 2.8.1. Safety Data

Serious and unexpected adverse events, such as unexpected deaths (i.e., not related to the progression of the primary disease or to limitation of care), severe hypercapnic acidosis that may be related to the intervention (arterial pH < 7.15, in the absence of metabolic acidosis and despite further tidal volume and/or respiratory rate increase) or the development of malignant hyperthermia, propofol-related infusion syndrome or bronchopleural fistula persistent despite drainage, will be reported to the sponsor by the investigators.

#### 2.8.2. Data and Safety Monitoring

Interim safety reports are performed by the sponsor to the independent data monitoring and safety committee (DMSC), consisting of two clinician-scientists and a methodologist-biostatistician, each time 40 patients are enrolled [14], including blinded variables on randomization-stratification variables (site, severe ARDS, COVID-19, and shock at enrollment), serious and unexpected adverse events, and the rates and causes of death at day 28. An interim analysis is preplanned on data from 350 patients (175 by group), with symmetric group sequential flexible stopping boundaries [41]. Recommendations for continuing, pausing, or stopping the study will be made by the DMSC to the steering committee for superiority of either active or control, or potential safety reasons. The observation of differences in serious adverse events between the two groups allows unblinding of the group allocations if deemed necessary by the DMSC.

#### 2.8.3. Statistical Considerations

The statistical analysis plan is provided in Appendix A. A two-sided *p* value < 0.05 will be considered for significance of all analyses, except for interim analysis. The primary analysis will be performed using an intention-to-treat (ITT) principle. Then, we will perform per-protocol and subgroup analyses on the primary and secondary outcomes (Table 3). The primary endpoint (VFD28) will be analyzed using a mixture of generalized gamma distributions to concatenate the overall frequency and distribution of the times [42]. Multivariable adjustments will be performed in (1) a first model including only the randomization-stratification variables and center as random-effect; (2) a second model with covariates from the first model and covariates with clinically relevant relationships with the outcome and significant in univariate analyses (*p* < 0.10). The key secondary outcome of day 90 survival will be evaluated using the Kaplan–Meier approach and compared using the log-rank test (univariate analysis); multivariable adjustments will be conducted using the same two models described above. Because of the potential for type I error due to multiple comparisons, findings for analyses of other secondary endpoints will be interpreted as exploratory.

#### 2.8.4. Sample-Size Justification

Total enrollment will target 700 patients (350 by group). The trial is designed with >80% power to detect a difference of 2 days alive and off ventilation at day 28 (primary outcome of VFD28), with standard deviation of 8 [43,44] and two-sided type I error of 5%. For this calculation, we assumed the variability of VFD28 would follow the properties of ROSE [21], the Esophageal Pressure-guided Ventilation-2 (EPVent-2) trial [45], and a pilot study of inhaled sevoflurane in ARDS [14], with 28 day mortality around 30–35% [21,24,45].

## 3. Discussion

The SESAR trial was designed to investigate the efficacy and safety of inhaled sevoflurane compared with intravenous propofol in patients with ARDS. The trial uses prognostic enrichment by enrolling the most severe patients (i.e., with a PaO_2_/FiO_2_ <150 mmHg) and, therefore, a greater probability of experiencing worse clinical outcomes [46,47]. The protocol implies strict adherence to evidence-based guidelines for analgesia-sedation and ventilation in ARDS, including recommendations on the ABCDEF bundle for ICU liberation and avoidance of benzodiazepines [48,49], and on lung-protective ventilation with low tidal volume and PEEP, prone position, and conservative fluid therapy unless the patient is in shock [27,50,51]. Based on the recent Severe ARDS: Generating Evidence (SAGE) Study, a multicenter, observational cohort study of 2466 mechanically ventilated adult patients with ARDS and PaO_2_/FiO_2_ < 150 mmHg on PEEP > 5 cmH_2_O in the United States, only adherence to early lung protective ventilation was associated with lower mortality [52]. The SESAR protocol stresses the use of lower tidal volumes (4 to 8 mL/kg PBW), and data monitoring has been developed to reveal whether higher tidal volumes are applied.

Early deep sedation will be targeted and associated with a systematic neuromuscular blockade in the trial, which is debatable [18,20,21]. However, this overall trial strategy has been chosen to avoid the substantial center-to-center variability that exists in ARDS management, while ensuring implementation of underused evidence-based interventions such as lung-protective ventilation and prone position [52]. In both study arms, deep sedation followed by a neuromuscular blockade will be initiated to reduce the risk of patient–ventilator asynchrony and patient self-inflicted lung injury [53]. This initial strategy will be switched for a lighter or no-sedation strategy as soon as oxygenation improves, thus better personalizing sedation and the neuromuscular blockade [24]. After this period, priority will be given to respiratory drive control and patient–ventilator synchrony to avoid the unnecessary use of medications such as sedatives, opioids, and neuromuscular blocking agents (paralysis over 48 h after randomization will be considered a rescue therapy) [18]. PEEP will be individually titrated based on plateau pressure, regardless of its effect on oxygenation in contrast to the PEEP/FiO_2_ scales used in other studies [32,54].

The trial will evaluate VFD28 as a primary, patient-centered outcome measure that represents the number of days alive and free of invasive mechanical ventilation calculated over 28 days. This endpoint, designed to assign benefit to early liberation from mechanical ventilation while considering death as a competitive risk, has been approved by the US Food and Drug Administration for licensing and is a clinically relevant outcome for phase III ARDS trials enrolling more than 500 patients [46]. Longer-term outcomes will also be evaluated in the trial, as there is an increased recognition of the significant long-term consequences of ARDS [55,56].

The precise pathways to lung-protective effects of sevoflurane, as observed in preclinical models and a pilot clinical trial [3,11,14], are largely unknown. The most commonly proposed mechanism is through the preservation of alveolar-epithelial integrity [57] and a reduction in pro-inflammatory cytokine release [6,7,9,10,14,58]. The SESAR trial includes the rigorous and sequential collection of plasma, alveolar, and urine samples that will be used to better characterize the effects of sevoflurane on major pathophysiological features of ARDS, in addition to assess renal laboratory markers and levels of sevoflurane metabolites. The results will also inform whether specific ARDS phenotypes might better benefit from inhaled sevoflurane, and which clinico-biological features and/or natural histories may be capable of predicting therapeutic response, and not just prognosis [59,60,61,62].

Although routinely used in some countries, inhaled sedation practice may largely vary among ICUs [2,4]. Volatile anesthetics use for ICU sedation requires specific training and specialized equipment, and some specific settings have also been suggested to increase viral protection in COVID-19 patients while ensuring safety to healthcare workers [63]. Sevoflurane is a well-known trigger to malignant hyperthermia in genetically predisposed patients, a syndrome that requires early intervention with the immediate change of the ventilator circuit and dantrolene infusion. However, this condition remains rare (1/50,000–100,000) in comparison to propofol infusion syndrome, which affects up to 1% of ICU patients [64]. The trial includes the remote and on-site training of all clinical teams to ensure strict adherence to the intended use and safety of inhaled sedation. Because we lack a cost-effectiveness analysis that takes into account any beneficial clinical outcomes such as faster awakening, extubation times, and lengths of ICU stay [65,66], the trial will also explore cost-effectiveness of sevoflurane compared with current practices.

SESAR is a randomized multicenter trial of inhaled sevoflurane in patients with ARDS. If the trial results demonstrate that sevoflurane increases VFD28 (primary endpoint) or 90 day survival (key secondary endpoint), compared with the current practice of intravenous propofol, increased use of inhaled ICU sedation should be considered. Better understanding of how sevoflurane may target major biological features of lung alveolar injury and their contributions to heterogeneity of the treatment effect will enhance the interpretation of trial results and inform future efforts in the early treatment of ARDS.

## Figures and Tables

**Figure 1 jcm-11-02796-f001:**
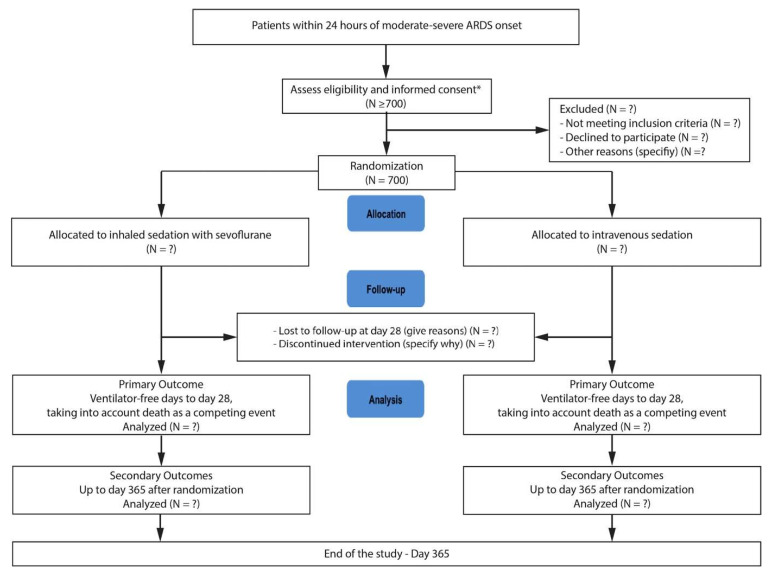
CONSORT diagram of the SESAR trial. * Because, in emergency situations, sedation and ventilation must be initiated as early as possible, the study protocol provides for a waiver of informed consent from the patient. The consent from the patient’s next of kin will therefore be sought actively during the 24 h enrollment time window. In case the patient’s next of kin cannot be reached in a timely manner, the investigator will decide to include the patient in the study using an emergent consent procedure. Deferred informed consent will be obtained from participants for potential continuation of the research.

**Figure 2 jcm-11-02796-f002:**
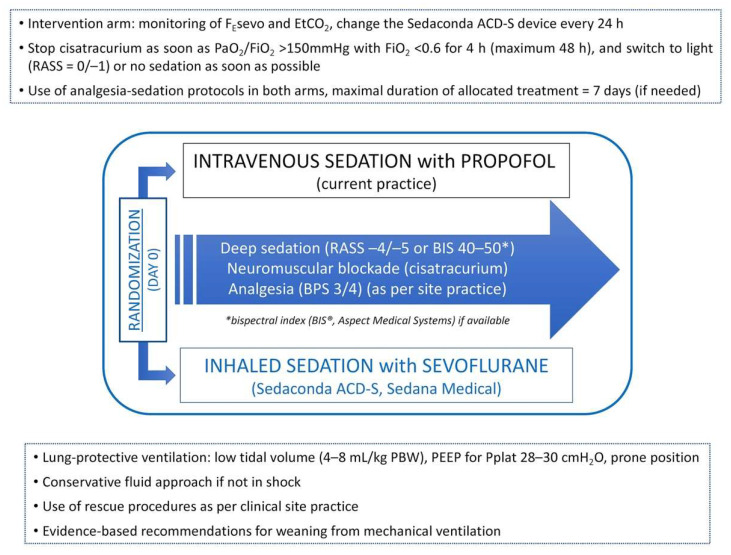
Summary of interventions within the SESAR trial. Definition of abbreviations: SESAR = sevoflurane for sedation in ARDS; ARDS = acute respiratory distress syndrome; RASS = Richmond agitation-sedation scale; BPS = behavioral pain scale; PBW = predicted body weight; PEEP = positive end-expiratory pressure; Pplat = inspiratory plateau pressure.

**Table 1 jcm-11-02796-t001:** Inclusion and exclusion criteria.

Inclusion criteria	Age ≥ 18 years Presence for ≤24 h of all the following conditions, within one week of a clinical insult or new or worsening respiratory symptoms: a. PaO_2_/FiO_2_ < 150 mmHg with positive end-expiratory pressure (PEEP) ≥8 cmH_2_O ^i,ii,iii^ or, if arterial blood gas is not available, SpO_2_/FiO_2_ ratio that is equivalent to a PaO_2_/FiO_2_ <150 mmHg with PEEP ≥ 8 cmH_2_O, and a confirmatory SpO_2_/FiO_2_ ratio between 1–6 h after the initial SpO_2_/FiO_2_ ratio determination ^iii,iv^ b. Bilateral opacities not fully explained by effusions, lobar/lung collapse, or nodules c. Respiratory failure not fully explained by cardiac failure or fluid overload; needs objective assessment (e.g., echocardiography) to exclude hydrostatic edema if no risk factor is present d. The 24 h enrollment time window begins when criteria a–c are met.
Exclusion criteria	Absence of affiliation to the French *Sécurité Sociale* Patient under a tutelage measure or placed under judicial protection Continuous sedation with inhaled sevoflurane at enrollment Known pregnancy Currently receiving ECMO therapy Chronic respiratory failure defined as PaCO_2_ >60 mmHg in the outpatient setting Home mechanical ventilation (non-invasive ventilation or via tracheotomy) except for CPAP/BIPAP used solely for sleep-disordered breathing Body mass index >40 kg/m^2^ Chronic liver disease defined as a Child–Pugh score of 12–15 Expected duration of mechanical ventilation <48 h Moribund patient, i.e., not expected to survive 24 h despite intensive care Burns > 70% total body surface Previous hypersensitivity or anaphylactic reaction to sevoflurane or cisatracurium Medical history of malignant hyperthermia Long QT syndrome at risk of arrhythmic events Medical history of liver disease attributed to previous exposure to a halogenated agent (including sevoflurane) Known hypersensitivity to propofol or any of its components Known allergy to eggs, egg products, soybeans, and soy products Suspected or proven intracranial hypertension Tidal volume of 6 mL/kg PBW below 200 mL (as recommended by the manufacturer for the use of the Sedaconda ACD-S (Sedana Medical, Danderyd, Sweden) Enrollment in another interventional ARDS trial with direct impact on sedation and mechanical ventilation Endotracheal ventilation for greater than 120 h (5 days) Persistent bronchopleural fistula despite chest tube drainage PaO_2_/FiO_2_ (if available) >200 mmHg after meeting inclusion criteria and before randomization.

*Definition of abbreviations:* PaO_2_ = partial pressure of arterial oxygen; FiO_2_ = fraction of inspired oxygen; PEEP = positive end-expiratory pressure; ECMO = extracorporeal membrane oxygenation; PaCO_2_ = partial pressure of arterial carbon dioxide; CPAP = continuous positive airway pressure; BIPAP = bi-level positive airway pressure; PBW = predicted body weight; ARDS = acute respiratory distress syndrome. i. If altitude > 1000 m, then PaO_2_/FiO_2_ < 150 x (PB/760). ii. These inclusion criteria ensure a non-transient, established hypoxia that persists despite elevated PEEP and time. Initial, post-intubation, PEEP is typically <8 cmH_2_O. iii. The qualifying PaO_2_/FiO_2_ or the SpO_2_/FiO_2_ must be from intubated patients receiving at least 8 cmH_2_O PEEP. iv. When hypoxia is documented using pulse oximetry, a confirmatory SpO_2_/FiO_2_ ratio is required to further establish persistent hypoxia. Qualifying SpO_2_/FiO_2_ must use SpO_2_ values less than or equal to 96% Qualifying SpO_2_ must be measured at least 10 min after any change to FiO_2_. The first qualifying SpO_2_/FiO_2_ (not the confirmatory SpO_2_/FiO_2_) is used to determine the 24 h enrollment time window. See Appendix A for details on imputations of PaO_2_/FiO_2_ based on combinations of SpO_2_ and FiO_2_.

**Table 2 jcm-11-02796-t002:** SESAR trial outcome measures.

Primary outcome	Ventilator-free days through day 28 (VFD28), as defined as the number of days alive and off the ventilator at 28 days, thereby considering death as a competing event *
Key secondary outcome	90 day survival (assessed on study day 91)
Secondary outcomes	All-location, all-cause 28 day mortality (assessed on study day 29) All-cause hospital 28 day mortality (assessed on study day 29) All-location, all-cause 14 day mortality (assessed on study day 15) All-location, all-cause 7 day mortality (assessed on study day 8)
Exploratory outcomes	Ventilator-free days through day 14 (VFD14) Ventilator-free days through day 7 (VFD7) Organ failure-free days through day 7 ** ICU-free days through day 28 Hospital-free days through day 28 Physiological measures to include: o Changes in oxygenation index, PaO_2_/FiO_2_, PaCO_2_, and arterial pH from day 1 to day 7 (defined as continuous time-dependent variables) o Changes in the level of PEEP (and static auto-PEEP in patients under controlled ventilation), inspiratory plateau pressure and static compliance of the respiratory system, and in ventilatory ratio^#^ from day 1 to day 7 (defined as a continuous time-dependent variable) Use of rescue procedures for refractory hypoxemia through day 28: nitric oxide, epoprostenol sodium, high frequency ventilation, ECMO, and neuromuscular blockade use after 48 h from randomization. ICU-acquired delirium: CAM-ICU assessed daily from study entry to study day 7, death or ICU discharge, whichever comes first. Long-term outcome assessments at 3 and 12 months: o Disability: Katz Activities of Daily Living o Health-Related Quality of Life: Short Form-36 o Pain-interference: 1 standard item o Post-traumatic Stress-like Symptoms: Post-Traumatic Stress Symptoms-14, Hospital Anxiety and Depression Scale o Cognitive function: Alzheimer’s Disease 8 o Subsequent return to work, hospital and ED use, and location of residence Healthcare-related costs during ICU stay and hospital stay
Exploratory biological outcomes	Change in plasma biomarkers of IL-8, sTNFr1, bicarbonates (hyperinflammatory ARDS phenotype), IL-6 (VILI), ANG-2 (endothelial activation), and sRAGE (alveolar epithelial injury) (defined as continuous time-dependent variables) ^§^ Change in urine biomarkers of TIMP-2 and IGFBP-7 (acute kidney injury) ^§^ Change in plasma total fluoride and hexafluoroisopropanol (sevoflurane metabolism) Genetic analysis: DNA and RNA at baseline and 48 h Change in total protein within undiluted pulmonary edema fluid at baseline and 24 h (alveolar fluid clearance) ^£^ Biomarker measurements in the fluid from the HME filter (control group) and Sedaconda ACD-S device (intervention group) at baseline and 24 h ^$^ Biomarker measurements in the BAL fluid within 48 h from study entry and between day 4 and day 6 ^€^
Safety outcomes	Changes in hemodynamic measures (mean arterial pressure, dose of infused norepinephrine or other vasopressor, serum lactate level) and in KDIGO criteria for acute kidney injury from day 1 to day 7 (defined as continuous time-dependent variables) Supraventricular tachycardia or new onset atrial fibrillation through day 7 Severe hypercapnic acidosis with arterial arterial pH < 7.15 ^%^ through day 7 (Sedaconda ACD-S device) Development of malignant hyperthermia through day 7 (sevoflurane) Development of propofol-related infusion syndrome through day 7 (propofol) Development of pneumothorax or bronchopleural fistula persistent despite drainage, through day 7

*Definition of abbreviations:* SESAR = sevoflurane for sedation in acute respiratory distress syndrome; VFD = ventilator-free days; ICU = intensive care unit; PEEP = positive end-expiratory pressure; ECMO = extracorporeal membrane oxygenation; CAM-ICU = confusion assessment method for the ICU; ED = emergency department; IL-8 = interleukin 8; sTNFR1 = soluble tumor necrosis factor receptor 1; IL-6 = interleukin 6; ANG-2 = angiopoietin 2; sRAGE = soluble receptor for advanced glycation end-products; TIMP-2 = tissue inhibitor of metalloproteinase 2; IGFBP-7 = insulin-like growth factor binding protein 7; DNA = deoxyribonucleic acid; RNA = ribonucleic acid; HME = heat moisture exchanger; BAL = bronchoalveolar lavage; KDIGO = kidney disease improving global outcomes. * Ventilator-free days through day 28 are defined as the number of days from the time of initiating unassisted breathing to day 28 after randomization, assuming survival for at least two consecutive calendar days after initiating unassisted breathing and continued unassisted breathing to day 28. If a patient returns to assisted breathing and subsequently achieves unassisted breathing to day 28, VFDs will be counted from the end of the last period of assisted breathing to day 28. A period of assisted breathing lasting less than 24 h and for the purpose of a surgical procedure will not count against the VFD calculation. If a patient was receiving assisted breathing at day 27 or dies prior to day 28, VFDs will be zero. Patients transferred to another hospital or other healthcare facility will be followed to day 28 to assess this endpoint. ** Organ failure is defined as present on any date when the most abnormal vital signs or clinically available lab value meets the definition of clinically significant organ failure according to SOFA scores. Patients will be followed for the development of organ failures to death, hospital discharge, or study day 7, whichever comes first. Each day a patient is alive and free of a given organ failure will be scored as a failure-free day. Any day that a patient is alive and free of all organ failures will represent days alive and free of all organ failure. # Ventilatory ratio = [minute ventilation (mL/min) × PaCO_2_ (mmHg)]/[predicted body weight × 100 × 37.5] ^§^ Plasma and urine samples will be collected from indwelling catheters (when available) at study entry and on days 1, 2, 4, 6, and 14 or ICU discharge (whichever occurs first). ^£^ In 50 patients from each group. ^$^ In 30 patients from each group. ^€^ In a total of 25 patients. ^%^ In the absence of metabolic acidosis and despite further tidal volume and/or respiratory rate increase, as described in the protocol.

**Table 3 jcm-11-02796-t003:** SESAR trial populations for primary, key secondary, secondary, and subgroup analyses.

Intention-to-treat population	All randomized patients except those who withdraw their consent for the use of data.
Per-protocol population #1	All randomized patients except patients having one or more major protocol violations defined as: Inhaled sevoflurane not administered in patients randomly allocated to the intervention arm *OR* 2. Inhaled sevoflurane not administered during the whole duration of sedation (within a maximum of 7 days from randomization) in patients randomly allocated to the intervention arm *OR* 3. Monitoring revealed that a tidal volume higher than 8 mL/kg PBW was applied *OR* 4. Monitoring revealed that one or more inclusion or exclusion criteria were violated *OR* 5. Patients withdrawn from the protocol because the patient would have withdrawn consent
Per-protocol population #2	All randomized patients except patients having one or more major protocol violations defined as: Inhaled sevoflurane was not administered in patients randomly allocated to the intervention arm *OR* 2. Inhaled sevoflurane was not administered during the whole duration of sedation (within a maximum of 7 days from randomization) in patients randomly allocated to the intervention arm
Subgroup populations	Patients with shock (defined as the need for intravenous vasopressor infusion to maintain arterial pressure) at randomization Patients with pre-randomization PaO_2_/FiO_2_ <100 mmHg Pre-randomization presence vs. absence of suspected COVID-19 Patients with hypoinflammatory versus hyperinflammatory phenotypes at randomization Patients with higher versus lower degrees of lung epithelial injury at randomization, as assessed by plasma sRAGE (thresholds to be determined according to univariate analyses and clinical relevance) Patients with higher versus lower degrees of lung endothelial injury at randomization, as assessed by baseline plasma ANG-2 (thresholds to be determined according to univariate analyses and clinical relevance) Patients with focal versus nonfocal ARDS at baseline, as assessed by lung CT-scan, chest radiograph, or bedside lung ultrasound (if available) Patients treated with lower (5–10 cmH_2_O), moderate (11–15 cmH_2_O), or higher (>15 cmH_2_O) levels of PEEP during the first 3 days after enrollment

Definition of abbreviations: SESAR = sevoflurane for sedation in acute respiratory distress syndrome; PBW = predicted body weight; sRAGE = soluble receptor for advanced glycation end-products; ANG-2 = angiopoietin 2; CT = computerized tomography; PEEP = positive end-expiratory pressure; COVID-19 = coronavirus disease 2019.

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
