# Peer review of "Design and Rationale of the Sevoflurane for Sedation in Acute Respiratory Distress Syndrome (SESAR) Randomized Controlled Trial"

_jcm, 2022, doi:10.3390/jcm11102796_

Round 1

Reviewer 1 Report

Summary

The authors design a phase 3 clinical trial to evaluate the efficacy of sedation with inhaled sevoflurane compared to intravenous propofol in patients with moderate-severe ARDS. The primary outcome is the number of days alive and off the ventilator at 28 days.

General concerns:

  1. The authors described the benefits of inhaled sevoflurane in improving pulmonary status of ARDS. Please describe the rationale to compare with intravenous propofol rather than intravenous midazolam in this clinical trial.
  2. Appendix 4_Protocol for weaning from mechanical ventilation: Patients can be assessed for weaning readiness criteria twice a day: Please correct “3. Values of both PEEP and FiO2 ≤values from previous day”.

Author Response

REVIEWER #1

The authors design a phase 3 clinical trial to evaluate the efficacy of sedation with inhaled sevoflurane compared to intravenous propofol in patients with moderate-severe ARDS. The primary outcome is the number of days alive and off the ventilator at 28 days.

General concerns:

C1. The authors described the benefits of inhaled sevoflurane in improving pulmonary status of ARDS. Please describe the rationale to compare with intravenous propofol rather than intravenous midazolam in this clinical trial.

R1. We thank Reviewer #1 for this important suggestion. Indeed, avoidance of benzodiazepines for sedation of ICU patients is currently recommended by the PADIS guidelines and should be emphasized.

Accordingly, we have added a mention in the first paragraph of the Discussion (page 11) in our revised manuscript: 

“The protocol implies strict adherence to evidence-based guidelines for analgesia-sedation and ventilation in ARDS, including recommendations on the ABCDEF bundle for ICU liberation and avoidance of benzodiazepines [49,50],...”

C2. Appendix 4_Protocol for weaning from mechanical ventilation: Patients can be assessed for weaning readiness criteria twice a day: Please correct “3. Values of both PEEP and FiO2 ≤values from previous day”.

R2. Thank you for pointing out this confusing sentence. We have changed the third criterion for weaning readiness in revised Appendix 4:

“3. Values of both PEEP and FiO2 ≤values from previous assessment day

Reviewer 2 Report

This is a very interesting study proposition

The study is challenging and I do hope that you succeed in completing it so that we will realize the place for inhaled anesthetics in the critical care setting. 

One study, that you did not add to your literature review is the study by Meiser et al which was recently published in Lancet, Respiratory Medicine. Although the investigators studied Isoflurane, they did succeed in showing non-inferiority for the inhaled anesthetics.

There is also one previous study that was initiated, but the results of which where not published by Soukup et al (Trials 2012) and it would be interesting to find out what happened with it

I have several questions that I did not find the answer to in the article or appendix.

How will patients in the sevoflurane who require either ECMO or High Frequency Ventilation or have sevo stopped for acidosis be analyzed (as at that point they will stop receiving sevoflurane)

I do not think it is realistic that any of the patients included in your study will not have an arterial line in place (certainly after the initial assessment) and therefore I would consider removing the whole inclusion criteria of SpO2/FiO2 (text, tables and references). If it is deemed important to keep this criteria than please define (to the best of my knowledge analysis was done for P/F of 200 and 300 mmHg/

Figure 1: There are two identical lines (Lost to follow-up at day 28) In the same figure: It should be stated that primary analysis is according to intention to treat.

What happens to patients who require sedation for more than 7 days on the sevoflurane group ? at that time they will be changed to IV (if I understand correctly Figure 2). This will impact on the primary outcome of all patients who are not weaned from sedation by day 7. How will this be reconciled?

In section 2.6 you state that reviewers of outcome will be blinded to patient allocation. Is this possible, will they not review patient charts from which it will be obvious which regimen was used.

section 2.8.3 the authors refrain from taking into account multiple comparisons with the risk for type I error. In the statistical appendix (7) this is explained clearly "Because of the potential for type I error due to multiple comparisons, findings for analyses of secondary endpoints will be interpreted as exploratory and systematic correction of type I error will not be applied; analyses will focus not only on statistical significance but also on the magnitude of differences " I think some of this should be explained also in the paper itself.

Author Response

REVIEWER #2

This is a very interesting study proposition. The study is challenging and I do hope that you succeed in completing it so that we will realize the place for inhaled anesthetics in the critical care setting.

C3. One study, that you did not add to your literature review is the study by Meiser et al which was recently published in Lancet, Respiratory Medicine. Although the investigators studied Isoflurane, they did succeed in showing non-inferiority for the inhaled anesthetics.

R3. We sincerely thank Reviewer #2 for the supporting words concerning our study. We agree with the fact that the SED-001 trial from Meiser et al. (doi:10.1016/S2213-2600(21)00323-4) is a major piece of literature on inhaled sedation, as it is both the largest trial on isoflurane available to date and the landmark trial that recently allowed the official labeling of isoflurane for ICU sedation in many European countries. It is a shame we did not refer to this article in our previous version, and we have now added it to the first paragraph of the revised Introduction (page 4):

“Volatile anesthetics such as isoflurane or sevoflurane have long been used to provide general anesthesia in the operating room; however, inhaled sedation is emerging as an option to provide intensive care unit (ICU) sedation and a phase III multicenter randomized controlled trial found that isoflurane is non-inferior to propofol in maintaining targeted sedation levels in critically ill adult patients [1–5].”

C4. There is also one previous study that was initiated, but the results of which where not published by Soukup et al (Trials 2012) and it would be interesting to find out what happened with it

R4. We fully agree that such data would be invaluable if available.

C5. I have several questions that I did not find the answer to in the article or appendix.

How will patients in the sevoflurane who require either ECMO or High Frequency Ventilation or have sevo stopped for acidosis be analyzed (as at that point they will stop receiving sevoflurane)

R5. Thank you for this practical -yet important- question.

These patients will be analyzed:

  • according to their primary randomization group in the Intention-To-Treat analysis,
  • within Per-Protocol populations #1 & #2 if inhaled sevoflurane is not administered (whatever the reason) during the whole duration of sedation (within a maximum of 7 days from randomization) in patients randomly allocated to the intervention arm (see Table 3 - page 28).

Patients will also be considered as receiving a “rescue therapy for refractory hypoxemia” in case they require ECMO or other rescue interventions such as listed in the protocol.

C6. I do not think it is realistic that any of the patients included in your study will not have an arterial line in place (certainly after the initial assessment) and therefore I would consider removing the whole inclusion criteria of SpO2/FiO2 (text, tables and references). If it is deemed important to keep this criteria than please define (to the best of my knowledge analysis was done for P/F of 200 and 300 mmHg.

R6. We fully agree with Reviewer #2 that SpO2/FiO2 might rarely be used for enrollment in our trial, because eligible patients are very likely to be equipped with an arterial line, thus providing easier access to PaO2/FiO2 assessments.

However, this alternative was initially included to facilitate the early screening of patients fulfilling the inclusion criteria even before an arterial line is in place.

In our revised manuscript (legends of Table 1, page 25), we have now added a mention to further details on how to impute PaO2/FiO2 based on combinations of SpO2 and FiO2, as initially described in the trial protocol:

“See Appendix S1 for details on imputations of PaO2/FiO2 based on combinations of SpO2 and FiO2.”

C7. Figure 1: There are two identical lines (Lost to follow-up at day 28) In the same figure: It should be stated that primary analysis is according to intention to treat.

R7. Thank you very much for pointing out this mistake. This has been fixed in revised Figure 1.

C8. What happens to patients who require sedation for more than 7 days on the sevoflurane group ? at that time they will be changed to IV (if I understand correctly Figure 2). This will impact on the primary outcome of all patients who are not weaned from sedation by day 7. How will this be reconciled?

R8. As described in the “Common strategies for both groups” subsection of the “MATERIALS AND METHODS” section (page 7), decisions on sedation after day 7 will be as per the treating clinicians. We acknowledge this implies that patients randomized to sevoflurane might then receive propofol (and vice versa), which can be a source of bias. This was, however, a requirement from the French Medicine Agency (Agence Nationale de Sécurité du Médicament et des Produits de Santé) to determine a maximum duration for sedation with the intervention drug sevoflurane as provided by the study sponsor within the trial.

We believe that such a duration (of 7 days) should be enough to associate with potential benefits, if any, in this population, although this is of course very debatable. Further sensitivity analyses will also be useful to assess whether the duration of inhaled sedation with sevoflurane has an impact on clinical outcomes.

C9. In section 2.6 you state that reviewers of outcome will be blinded to patient allocation. Is this possible, will they not review patient charts from which it will be obvious which regimen was used.

R9. Theoretically, our eCRF allows blinded data entry by trained clinical research associates, who are not part of the clinical teams taking care of the patients; they should, therefore, remain blinded to the allocation group, at least in theory. More importantly, outcome assessment and statistical analyses will be performed in a blinded manner.

C10. section 2.8.3 the authors refrain from taking into account multiple comparisons with the risk for type I error. In the statistical appendix (7) this is explained clearly "Because of the potential for type I error due to multiple comparisons, findings for analyses of secondary endpoints will be interpreted as exploratory and systematic correction of type I error will not be applied; analyses will focus not only on statistical significance but also on the magnitude of differences " I think some of this should be explained also in the paper itself.

R10. We agree with Reviewer #2 that this is an important point, and have added a mention to it at the end of the “Statistical Considerations” subsection (page 11):

Because of the potential for type I error due to multiple comparisons, findings for analyses of other secondary endpoints will be interpreted as exploratory.

This manuscript is a resubmission of an earlier submission. The following is a list of the peer review reports and author responses from that submission.